# Patient Safety Related to Microbiological Contamination of the Environment of a Multi-Profile Clinical Hospital

**DOI:** 10.3390/ijerph18073844

**Published:** 2021-04-06

**Authors:** Marlena Robakowska, Marek Bronk, Anna Tyrańska-Fobke, Daniel Ślęzak, Jakub Kraszewski, Łukasz Balwicki

**Affiliations:** 1Department of Public Health & Social Medicine, Medical University of Gdańsk, 80-210 Gdańsk, Poland; marlena.robakowska@gumed.edu.pl (M.R.); balwicki@gumed.edu.pl (Ł.B.); 2Laboratory of Clinical Microbiology, University Center of Laboratory Medicine, University Clinical Center in Gdańsk, 80-952 Gdańsk, Poland; msb@gumed.edu.pl; 3Department of Medical Rescue, Medical University of Gdańsk, 80-210 Gdańsk, Poland; 4University Clinical Center in Gdańsk, 80-952 Gdańsk, Poland; jkraszewski@uck.gda.pl

**Keywords:** infection prevention and control, surveillance of infections, epidemiology of infections

## Abstract

Nosocomial infections pose a serious burden for hospitals, patients, and the entire society. The aim of the study was to assess the microbiological cleanliness of the hospital environment through quantitative and qualitative analysis of microbiological contamination of air and surfaces in inpatient treatment facilities, based on the example of a large clinical hospital in Poland. Data were collected between 2012 and 2018 in premises of a large teaching hospital in Gdansk using the sedimentation method and the impact method using the Aerideal apparatus (Biomerieux). In the analyzed clinical center, the microbiological cleanliness tests in most of the hospital rooms in the analyzed period showed an acceptable number of saprophytic microorganisms. Of all the tested samples, 1159 (21.8%) were positive, indicating the presence of microorganisms in the tested sample. Species potentially pathogenic for hospital patients were identified, constituting 20.8% of all positive samples (4.6% of all samples). Significantly higher proportion of microorganisms potentially dangerous to patients were isolated from sanitary facilities. Due to the potentially pathogenic microorganisms detected in the tested samples, the authors suggest that in the analyzed hospital, the areas requiring a specific level of microbiological purity should be designated and described, with [specifically] defined cleaning and disinfection protocols.

## 1. Introduction

Nosocomial infections are inherently associated with patient stay in a hospital or another facility providing medical services 24 h a day. Hospitalized people, who often have a reduced immune resistance at the time of their admission to hospital, undergo various types of treatments, and consequently are more susceptible to infections than healthy people. The WHO defines an adverse event, which includes nosocomial infections, as damage caused during or as a result of treatment and not related to the natural course of the disease or the patient’s state of health. It can occur at any stage of hospitalization (i.e., from the moment of admitting the patient to the hospital until their discharge) [1].

Paradoxically, the growth in the importance of the problem of nosocomial infections correlates positively with medical advances. The widespread use of modern treatment techniques (e.g., endoscopy, vascular cannulation, assisted respiration, organ transplants, etc.) has resulted in the opening of numerous “gateways” for infection by microorganisms among patients treated in hospital settings. Moreover, the widespread and not always controlled use of antibiotics has become one of the reasons for the rise in antimicrobial resistance in hospital environments [2]. This has led to the development of so-called “hospital strains”, which are resistant to most known and used antibacterial drugs. Another very important reason for the increase in the number of nosocomial infections is imperfections and sometimes neglect in hospital maintenance [3]. Surfaces in the hospital environment are often seriously contaminated [4].

Nosocomial infections place a heavy burden on hospitals, patients, and society at large. One British study found that a hospital-acquired infection prolongs the stay of a patients in hospital by 250%, nearly tripling the cost of hospital treatment, while after discharge, such patients require more frequent services from family doctors and nurses and higher drug spending than patients without nosocomial infections [3].

In Poland, a classification of hospital rooms according to the required microbiological air purity was introduced in 1984 in order to ensure the proper level of hygiene in hospital rooms, depending on their purpose and method of use. In the Polish standard, three classes of room cleanliness are defined based on the acceptable concentration of bacteria in the air. Current standards and guidelines in [other] European countries base the classification of [hospital] rooms on the quantitative concentration of all microorganisms.

Another Polish standard, DIN 1946-4: 2008-12, in accordance with the recommendations of the Robert Koch Institute, classifies hospital rooms into two classes of microbiological cleanliness: I and II. Class I is further divided into two subclasses: Ia and Ib.

Microbiological tests of the hospital environment are recommended in the event of an epidemic outbreak, if the environment is the most likely source of an epidemically spreading microorganism [2,3,5]. On the other hand, routine microbiological testing of the environment is controversial, mainly due to high financial costs and the view that testing of surfaces using the smear method is characterized by low and variable sensitivity [6,7]. In recent years, however, solutions have been sought to enable bacteriological assessment of surface cleanliness, following the example of the standards applicable, for example, in the food industry. These proposed testing standards include two elements: the presence of the so-called indicator microorganisms and an assessment of the total number of microorganisms. Indicator microorganisms are pathogens, the presence of which is associated with a high risk of infection in hospitalized people. Among others, *S. aureus* strains including MRSA (methicyllin-resistant Staphylococcus aureus), VRE (Vancomycin-Resistant Enterococcus), C. difficile, and multi-resistant Gram-negative rods [8,9,10,11,12] were considered indicator microorganisms. The total microbial count, on the other hand, is the basis for assessing the risk of surface pathogen transmission to the patient. The currently proposed standard for the acceptable surface contamination stipulate below 1 CFU/cm^2^ of indicator microorganisms and a maximum of 2.5 to 5 CFU/cm^2^ of the total number of microorganisms on touch surfaces [13]. The above interpretation of the surface microbiological test results does not take into account the different microbiological cleanliness requirements in hospital rooms depending on the patient’s exposure to microorganisms (e.g., operating rooms) or their susceptibility to infection (e.g., immunosuppressed patient rooms). Moreover, indicator microorganisms were selected from the group of bacterial pathogens, which is related to research methodology based mainly on the identification of aerobic bacteria.

Based on international experience, it can be expected that reducing the nationwide frequency of nosocomial infections in Poland by 1% may reduce the costs of hospital treatment by approximately 7% [9]. On an annual country-wide scale, this means 214 million USD in savings, just from the direct costs of hospitalization. The indirect costs associated with hospital deaths alone amount to approximately 118 million USD per year [9]. Treatment of infections where pharmacotherapy is necessary is extremely expensive. According to expert assessment, the cost of correctly conducted antibiotic therapy in the case of blood sepsis may range from 267 to 1337 USD. To this amount should be added, inter alia, the cost of extended hospitalization (not including other indirect costs) [12]. Costs that are not reimbursed by insurance companies or by the patient are borne by the hospital. Extension of the stay due to nosocomial infection of patients hospitalized for other reasons also means that the hospital cannot accept other patients. Despite the current level of medical knowledge, nosocomial infections still cannot be completely eliminated, however, the experience of countries that have been using surveillance and control programs for many years shows that with the help of such programs, the incidence of infection can be reduced [12].

The aim of the study was to assess the microbiological purity of the hospital environment through the quantitative and qualitative analysis of microbiological contamination of air and surfaces in inpatient treatment using the example of a large clinical hospital in Poland.

## 2. Materials and Method

The study was carried out between 2012 and 2018 in the hospital premises of a large teaching hospital in Gdansk. Swabs were collected from bathroom fixtures, beds, countertops, and bedding as well as monitors and devices in the department’s equipment. The material was collected randomly from patients’ rooms and treatment areas as well as staff rooms, bathrooms, toilets, and corridors. The material was collected once a week and was analyzed at the Clinical Microbiology Laboratory of the University Center for Laboratory Medicine, University Clinical Center in Gdańsk (Gdańsk, Poland).

The microbiological air purity tests were carried out using the sedimentation method and the collision method with the use of the Aerideal apparatus (Biomerieux, Warsaw, Poland). Columbia agar plates with 5% sheep blood (Biomerieux, Warsaw, Poland) and Sabouraud plates (Biomerieux, Warsaw, Poland) were used in the sedimentation method, which used the free fall of contaminated dust particles. In the test room, five plates were placed at the ground level, with four plates near the walls in the corners of the room and one tile in the center of the room. Each plate was exposed for one hour. Following exposure, the Columbia Sheep Blood Agar plates were incubated in an incubator at 36 °C under aerobic conditions for 48 h, then for 24 h at room temperature. After incubation, the colonies grown on all five plates were counted and the sum was divided by five to calculate the average number of colonies on the plate. Then, the number of contaminated particles in 1 m^3^ of air was calculated using the formula: X = a × 5 × 10^4^/πr^2^ × t,(1)
X—number of microorganisms in the air (in cfu/m^3^ or cfu/m^3^), a—number of colonies grown on a Petri dish, πr^2^—Petri dish surface area (in cm^2^) t—exposure time in minutes (in minutes).

The plates used for the collision sampling with the Aerideal apparatus were similar. In this case, the number of contaminated particles in 1 m^3^ of air was determined based on the number of colonies growing on the substrate in the plate and based on the data in the tables for the quantitative interpretation of the test results.

Microbes growing as colonies on solid agar media were identified by species or genus using biochemical tests performed on a Vitek automated species determination apparatus (Biomerieux, Warsaw, Poland).

The project analyzed 5730 samples of swabs collected from the clinics at the University Clinical Center in Gdańsk (Gdańsk, Poland). It is the largest academic and multi-profile inpatient health center in Pomerania, with a high level of service quality, supported by numerous certificates. The center, with nearly 1200 beds, employs nearly 4000 people, and about 120 thousand patients are treated annually, both from the Pomo’rskie voivodeship as well as from other areas of Poland.

A total of 5325 records containing complete data were qualified for further analysis. All analyzed samples were grouped according to the place of collection. The following areas were distinguished for the purpose of microbiological purity testing: hospital equipment, textiles, floors, hospital furniture, air, gloves, sanitary equipment, water, and “other”. The samples were then divided according to the level of cleanliness required of the places from which they were collected. Three categories were established based on the microbiological cleanliness classes for hospital rooms described above, in line with the RKI recommendations: A—no microbes; B—an acceptable small number of non-pathogenic (saprophytic) microorganisms; and C—an acceptable higher number of saprophytic microorganisms, but no microorganisms potentially pathogenic. The adopted classification concerns the presence of bacteria only. The samples in which the presence of fungi was detected were not included.

Statistical analysis of the significance of differences was carried out in order to investigate the relationship between selected groups of data. Using the Statistica statistical analysis software program (TIBCO Software Inc., Palo Alto, CA, USA) the sampling sites were analyzed in groups according to the cleanliness category and the types of microorganisms present in the examined hospital environment (*p* < 0.05).

## 3. Results

Of all the tested samples, 1159 (21.8%) cases were positive, meaning the presence of microorganisms in the tested sample. More than 30% of positive results were obtained in samples from sanitary facilities (*p* < 0.05). Over 30% of all collected samples came from hospital equipment. A detailed distribution of the obtained results in the studied areas is presented in Table 1.

Most of the collected samples belonged to category C, in other words, they showed an acceptable higher number of saprophytic microorganisms, but without potentially pathogenic microorganisms (39%) (*p* < 0.05, Pearson Chi-square 30, V Cramer 0.365). A detailed distribution of the bacteriological purity categories of the tested samples is presented in Table 2.

Among the positive results obtained, Gram-positive bacteria were isolated in 515 (44.5%) and Gram-strains were isolated in 475 (41%) samples (*p* < 0.05, Pearson Chi-square 8990, V Cramer 0.627). Fungi constituted 8% of microorganisms isolated from positive environmental swabs. Moreover, among the positive results, most of the bacteria isolated were aerobic bacteria on 576 swabs (49.7%) (*p* < 0.05). It should be noted here that more than one microorganism was isolated from the vast majority of positive samples, most often two or three different microorganisms. Then, the obtained positive results were analyzed in terms of the sampling site. A detailed distribution of the classification by types of the sampled samples is presented in Table 3 and Table 4. 

Bacteria identified in positive samples were also classified according to their families and genera. Among the analyzed positive samples, bacteria of the Staphylococcaceae family (31.9%) and the Staphylococcus genus (29.2%) constituted the largest group (*p* < 0.05). A detailed distribution of the families and types of bacteria in the samples is presented in Table 5 and Table 6.

It is true that among the obtained positive results indicating the presence of microorganisms in the tested samples, the vast majority were non-pathogenic microorganisms. However, in the studied hospital environment, in 241 cases, species potentially pathogenic for hospital patients were identified, constituting 20.8% of all positive samples and 4.6% of all analyzed samples. Significantly more microorganisms potentially dangerous to patients were isolated from sanitary facilities (*p* < 0.05). A detailed list of pathogenic microorganisms isolated during the entire study period is presented in Table 7.

## 4. Discussion

One of the factors influencing the incidence of nosocomial infections is the level of microbiological cleanliness of the patient’s environment. It has been shown that the environment of a medical facility can be a source of pathogens responsible for HAI (Healthcare Associated Infection) in nearly 20% of patients [14]. Moreover, it is estimated that the incidence of hospital-related infections in Poland is 5.9% [15]. In the analyzed clinical center, microbiological cleanliness testing in most of the hospital rooms in the analyzed period showed an acceptable higher number of saprophytic microorganisms, but without potentially pathogenic microorganisms. Pathogenic microorganisms were present in a small number of tested samples.

Maintaining appropriate cleanliness in the hospital environment is extremely important for the well-being of patients and medical staff. Patients with a weakened immune system staying in hospitals are very susceptible to infections, the source of which may be medical tools and equipment contaminated with microbes, equipment and building partitions in rooms, internal air, and other people staying in the hospital [2,3,4]. Many studies have shown the important role of the hospital environment in the transmission of pathogens, with infections occurring in patients placed in rooms where patients with infections previously stayed. This route of transmission has been proven for the following strains: *Staphylococcus aureus* resistant to methicillin (MRSA), *Endococcus faecium*, and *Enterococcus faecalis* resistant to vancomycin (VRE), *Pseudomonas aeruginosa*, *Acinetobacter baumannii*, and *Clostridium difficile* [16,17,18,19,20]. In the analyzed clinical center, only a negligible number of *Pseudomonas aeruginosa* and *Staphylococcus aureus* samples were detected. The other microorganisms listed above were not identified in this study.

A retrospective study of eight intensive care units (ICUs) in the United States, which analyzed 11,528 ICU stays, showed a significantly higher rate of MRSA (3.9% vs. 2.9%) and VRE (4.5% vs. 2.8%) among patients placed in rooms previously occupied by patients, infected or colonized by these pathogens [16]. The analysis of factors increasing the risk of VRE infection in 638 patients carried out in two intensive care units showed that placing the patient in the room where a patient previously stayed due to a VRE infection or colonization significantly increased the risk of infection with this pathogen [17]. In the study of infections caused by *P. aeruginosa* and *A. baumannii* resistant to multiple antibiotics in 511 ICU patients in France, an independent risk factor for infection was a stay in a room previously occupied by patients infected or colonized with these microorganisms [18]. During a two-month monitoring of the hospital environment in a hospital in the UK, a significant increase in the number of infected patients was shown when the number of bacteria present on surfaces in patient rooms exceeded 2.5 CFU/cm^2^ or when these surfaces were contaminated with *S. aureus* [8]. Studies on the survival of microorganisms on dry surfaces have shown that the survival time is variable and, for example, for Enterococcus spp. including VRE up to five days up to 46 months, *Staphylococcus aureus* up to 12 months, and *Clostridium difficile* up to five months. Bacteria such as Klebsiella spp. or *Pseudomonas aeruginosa* can survive for up to several months [4].

Similarly, in the intensive care unit for neurosurgery, a significant correlation was confirmed between environmental contamination and the frequency of colonization and infection of patients with the *A. baumannii* strain [21].

Undetected sources of infection or inadequate decontamination of the environment, especially of frequently touched surfaces, can cause persistent infection foci with multi-resistant *Acinetobacter baumannii* strains [21,22,23].

In one ICU in Argentina, improper decontamination of the room and bed of an infected patient led to the reactivation of the *A. baumannii* infection [23]. Improper decontamination of electronic equipment has been shown to play a significant role in the spread of infections with this etiology. The introduction of the disinfection of casings and disinfection of hands before and after contact with the keyboard can effectively prevent the spread of these bacteria in the hospital environment [24,25,26]. The change in the principles of decontamination of computer keyboards, monitor surfaces, and surfaces in the environment of patients infected with carbapenem-resistant *A. baumannii* strains allowed for the eradication of the epidemic strain from the environment and the extinguishing of an outbreak of infections caused by this microorganism in an ICU in Australia [25]. Improved cleaning in combination with microbiological control of the environment has also proved effective in controlling an outbreak of multi-resistant infections of this species, covering a total of more than 60 intensive care patients in Spain [22].

Unfortunately, the existing standards for acceptable concentrations of pollutants in the hospital environment still do not satisfy the needs resulting from the necessity of assessing the risks arising in the event of environmental contamination. In particular, there is a lack of commonly used, uniform standards for interpreting the results of quantitative and qualitative microbiological tests in the hospital environment. Most of the recommendations currently in use are adopted standards, developed mainly for the so-called clean rooms in industries with high requirements for dust and microbiological cleanliness of the surrounding environment (air and surface) such as, for example, the pharmaceutical, optical, and microelectronics industries, although operating rooms are also classified as clean rooms within the definitions used in these standards. In Poland, outdated guidelines are also used regarding the admissible microbiological concentrations in health care facilities, included in the “Guidelines for the design of general hospitals”.

Due to the lack of modern national air quality guidelines in health care facilities, following the global trends in terms of increasing air cleanliness requirements in the most critical hospital rooms, foreign guidelines are often used. The most widely known and applied standards and guidelines in Poland are the German ones. The current version of DIN 1946-4, published in December 2008, includes recommendations for both air-conditioning design (e.g., air parameters, required ventilation air flows) as well as a new method for assessing rooms with the highest air purity requirements. This method, which determines how well the relevant zone is protected against pollution from the environment, constitutes a completely new approach to the assessment of the correct operation and installation of air purification technology. The recommendations apply to the planning, construction, and classification of ventilation and air-conditioning systems in rooms used for examining patients and for performing procedures and operations (as well as any accompanying rooms). However, they do not apply to special treatment facilities for patients with highly infectious diseases.

## 5. Conclusions

Microbiological cleanliness of hospitals is one of key factors in preventing nosocomial infections. Our study indicates that mainly Gram-positive microorganisms with low pathogenic potential such as Micrococcus, Bacillus, and Staphylococcus (not *S. aureus*) can be found in the air and on dry surfaces. However, even those microorganisms may pose a threat to patients with severe immunodeficiency.

In a humid environment and on damp surfaces (sanitary equipment), Gram-negative bacteria of the genera Escherichia, Klebsiella, Pseudomonas, and Stenotrophomonas predominate. These can all cause infections in hospitalized patients. The potentially pathogenic species of *Staphylococcus aureus* is also present.

The presence of potentially pathogenic microorganisms indicate that in the hospital we studied, areas which require a specific level of microbiological purity should be designated and described, and specific protocols for cleaning and disinfection should be laid down for those areas. 

It would also be advisable, as a prophylactic measure implemented for the benefit of patients and in order to constantly improve the quality of hospital services, to develop a standard procedure for controlling and assessing environmental contamination in these areas. In order to comply with the proposed recommendations, systematic training and audits should be carried out. 

## Figures and Tables

**Table 1 ijerph-18-03844-t001:** Detailed distribution of the obtained results in the studied areas ^1^.

Study Area	Positive Result	Negative Result	No material Available	Overall
	N (%)	N (%)	N (%)	N (%)
Hospital equipment	235 (20.2%)	1370 (32.9%)	1 (50%)	1606 (30.2%)
Textile goods	46 (4%)	366 (8.8%)	0	412 (7.7%)
Floors	11 (1%)	56 (1.3%)	0	67 (1.3%)
Hospital furniture	157 (13.5%)	1200 (28.8%)	0	1357 (25.5%)
Air	323 (27.8%)	300 (7.2%)	1 (50%)	624 (11.7%)
Gloves	0 (0%)	7 (0.2%)	0	7 (0.1%)
Sanitary equipment	352 (30.4%)	751(18%)	0	1103 (20.7%)
Water	7 (0.6%)	11(0.3%)	0	18 (0.3%)
Other	28 (2.5%)	103 (2.5%)	0	131 (2.5%)
Total	1159	4166	2	5325

^1^*p* < 0.05.

**Table 2 ijerph-18-03844-t002:** Distribution of the bacteriological purity categories of the samples ^1^.

Study Area	Category A	Category B	Category C
	N (%)	N (%)	N (%)
Hospital Equipment	522 (31.2%)	404 (27.2%)	227 (10.9%)
Textile Goods	8 (0.5%)	243 (16.4%)	183 (8.8%)
Floors	37 (2.2%)	7 (0.5%)	30 (1.4%)
Hospital Furniture	527 (31.5%)	371 (25%)	480 (23.1%)
Air	274 (16.4%)	197 (13.3%)	170 (8.2%)
Gloves	7 (0.4%)	0	0
Sanitary Equipment	227 (13.6%)	245(16.5%)	902 (43.4%)
Water	1 (0.1%)	9 (0.6%)	11 (0.5%)
Other	68 (4.1%)	9 (0.6%)	74 (3.6%)
Total	1671	1485	2077

^1^*p* < 0.05.

**Table 3 ijerph-18-03844-t003:** Sampling site distribution by the type of the microorganism ^1^.

Study Area	Gram-positive	Gram-negative	Fungi
	N (%)	N (%)	N (%)
Hospital equipment	36 (7%)	7 (1.5%)	0
Textile goods	65 (12.6%)	2 (0.4%)	0
Floors	18 (3.5%)	0	0
Hospital furniture	162 (31.5%)	3 (0.6%)	60 (65.3%)
Air	33 (6.4%)	3 (0.6%)	0
Gloves	0	0	0
Sanitary equipment	160 (31.1%)	447 (94.1%)	20 (21.8%)
Water	2 (0.4%)	6 (1.3%)	0
Other	39 (7.6%)	7 (1.5%)	12 (13.1%)
Total	515	475	92

^1^*p* < 0.05.

**Table 4 ijerph-18-03844-t004:** Distribution of a sampling location category among areobes, anaerobes, and relative anaerobes ^1^.

Study Area	Aerobes	Anaerobes	Relative Anaerobes
	N (%)	N (%)	N (%)
Hospital equipment	22 (3.8%)	0	21 (5.1%)
Textile goods	28 (4.9%)	0	39 (9.5%)
Floors	10 (1.7%)	0	8 (1.95)
Hospital furniture	59 (10.2%)	0	106 (25.7%)
Air	20 (3.5%)	0	16 (3.9%)
Gloves	0	0	0
Sanitary equipment	418 (72.6%)	2 (100%)	187 (45.4%)
Water	6 (1%)	0	2 (0.5%)
Other	13 (2.3%)	0	33 (8%)
Overall	576	2	412

^1^*p* < 0.05.

**Table 5 ijerph-18-03844-t005:** Distribution of bacteria families by the sampling area ^1^.

Study Area	Hospital Equipment	Textile Goods	Floors	Hospital Furniture	Air	Gloves	Sanitary Equipment	Water	Other	Total
	N (%)	N (%)	N (%)	N (%)	N (%)	N (%)	N (%)	N (%)	N (%)	N
Alcaligenaceae	1 (6.3%)	0	0	0	0	0	15 (93.7%)	0	0	16
Bacillaceae	2 (4.9%)	10 (24.4%)	5 (12.2%)	11 (26.8%)	1(2.4%)	0	10 (24.4%)	0	2 (4.9%)	41
Clostridiaceae	0	0	0	0	0	0	1 (100%)	0	0	1
Corynebacteriaceae	0	0	0	0	0	0	5 (100%)	0	0	5
Enterobacteria	2 (1%)	1 (0.5%)	0	0	1(0.5%)	0	201(96.6%)	0	3(1.4%)	208
Micrococcaceae	14 (9.9%)	16 (11.3%)	5 (3.5%)	45 (31.9%)	14 (9.9%)	0	41 (29.1%)	0	6 (4.3%)	141
Moraxellaceae	1 (3.3%)	1 (3.3%)	0	0	0	0	24 (80%)	3(10%)	1 (3.3%)	30
Pseudomonandaceae	3 (2.1%)	0	0	2 (1.4%)	2 (1.4%)	0	131 (91%)	3 (2.1%)	3 (2.1%)	144
Staphylococcaceae	20 (6.3%)	39 (12.4%)	8 (2.5%)	106 (33.7%)	16 (5.1%)	0	93 (29.5%)	2 (0.6%)	31 (9.8%)	315
Streptococcaceae	0	0	0	0	2 (100%)	0	0	0	0	2
Xanthomonadaceae	0	0	0	1 (1.1%)	0	0	84 (98.9%)	0	0	85
Other	0	0	0	0	0	0	2 (100%)	0	0	2

^1^*p* < 0.05.

**Table 6 ijerph-18-03844-t006:** Distribution microorganism type by the sampling area ^1^.

Study Area	Hospital Equipment	Textile Goods	Floors	Hospital Furniture	Air	Gloves	Sanitary Equipment	Water	Other	Overall
	N (%)	N (%)	N (%)	N (%)	N (%)	N (%)	N (%)	N (%)	N (%)	N
Achromobacter	0	0	0	0	0	0	2 (100%)	0	0	2
Acinetobacter	1	1	0	0	0	0	24	3	1	30
Alcaligenes	1	0	0	0	0	0	15	0	0	16
Aspergilus	0	0	0	60	0	0	20	0	12	92
Bacillus	2 (4.9%)	10 (24.4%)	5 (12.2%)	11 (26.8%)	1(2.4%)	0	10 (24.4%)	0	2 (4.9%)	41
Citrobacter	0	0	0	0	0	0	8 (100%)	0	0	8
Clostridium	0	0	0	0	0	0	1 (100%)	0	0	1
Comomonas	0	0	0	0	0	0	1 (100%)	0	0	1
Corynebacterium	0	0	0	0	0	0	5 (100%)	0	0	5
Enterobacter	0	0	0	0	1	0	45	0	1	47
Enterococcus	0	0	0	0	0	0	10 (100%)	0	0	10
Escherichia	1 (1.3%)	0	0	0	0	0	73 (96%)	0	2 (2.7%)	76
Klebsiella	1 (1.8%)	1 (1.8%)	0	0	0	0	55 (96.4%)	0	0	57
Leconostoc	0	0	0	0	2 (100%)	0	0	0	0	2
Micrococcus	14 (9.9%)	16 (11.3%)	5 (3.5%)	45 (31.9%)	14 (9.9%)	0	41 (29.1%)	0	6 (4.3%)	141
Morganella	0	0	0	0	0	0	3 (100%)	0	0	3
Proteus	0	0	0	0	0	0	4 (100%)	0	0	4
Pseudomonas	3 (2.1%)	0	0	2 (1.4%)	2 (1.4%)	0	130 (90.9%)	3 (2.1%)	3 (2.1%)	143
Serratia	0	0	0	0	0	0	3 (100%)	0	0	3
Staphylococcus	20 (6.3%)	39 (12.4%)	8 (2.5%)	106 (33.7%)	16 (5.1%)	0	93 (29.5%)	2 (0.6%)	31 (9.8%)	315
Stenotrophomonas	0	0	0	1 (1.1%)	0	0	84 (98.9%)	0	0	85

^1^*p* < 0.05.

**Table 7 ijerph-18-03844-t007:** Pathogenic microorganisms isolated during the entire study period ^1^.

Study Area	Escherichia Coli	Klebsiella Pneumoniae	Pseudomonas Aeruginosa	Staphylococcus Aureus
Hospital equipment	N (%)	N (%)	N (%)	N (%)
Hospital	1(1.3%)	1 (2.5%)	2 (1.9%)	1 (5.6%)
Textile goods	0	0	0	2 (11.2%)
Floors	0	0	0	0
Hospital furniture	0	0	0	1(5.6%)
Air	0	0	0	0
Gloves	0	0	0	0
Sanitary equipment	75 (97.4%)	40 (97.5%)	98 (93.3%)	8 (44.5%)
Water	0	0	3 (2.9%)	0
Other	1 (1.3%)	0	2 (1.9%)	6 (33.1%)
Overall	77	41	105	18

^1^*p* < 0.05.

## Data Availability

Data sharing not applicable.

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
