# Peer review of "Patient Safety Related to Microbiological Contamination of the Environment of a Multi-Profile Clinical Hospital"

_ijerph, 2021, doi:10.3390/ijerph18073844_

Round 1

Reviewer 1 Report

The statistical analysis could be improved. Table3 is wrongly labelled as Table 5. Statistical software was quoted to have been used for the analysis but no presentation for the test of significance in the result. No p-value was presented in the tables but p<0.05 was put underneath the Tables. 

More statistical analysis could be done and it is desirable to improve the manuscript

Author Response

Point 1: The statistical analysis could be improved. Table3 is wrongly labelled as Table 5. Statistical software was quoted to have been used for the analysis but no presentation for the test of significance in the result. No p-value was presented in the tables but p<0.05 was put underneath the Tables.

Response 1: Thank you. Table 3 designation was changed to correct. Currently it is table 7. We added presentation for the test of significance in the result (line 157, 161, 168, 173, 175, 188).

Point 2: More statistical analysis could be done and it is desirable to improve the manuscript.

Response 2: Thank you. We have added the results of additional analyzes that present tables 3 to 6, i.e .:

Table 3. Sampling site distribution by the type of the microorganism.

Table 4. Distribution of a sampling location category among areobes, anaeobes and relative anaerobes.

Table 5. Distribution of bacteria families by the sampling area.

Table 6. Distribution microorganism type by the sampling area

Reviewer 2 Report

To help improve  readers interest to the work,  consider the following observsation; 

  1. The abstract need closer attention. In most cases, there are no punctuation marks where they are needed.  In addition it will be good to guide your reader around key results as well as the impleication of the study findings.  
  2. line 40-41 .......opened many "gates" for microorganisms.  The statement seem not to be complete. Are you reffering to their presence in the hosipital premises?
  3. Line 42-43  there is a jump.  In my opinion, you needed link their presence with resistence before the justification of their resistence made herein. 
  4. Check punctuation and the use of capital letters at the start of next sentence (i.e. line 53-54)
  5. line 109-111.  Considering the longitudinal nature of the study (2012-2018) it will have been much better if the authors detailed  the frequency of the sampling done i.e. once/wk , twice/wk, fortnightly etc  and what elevation were the plates and aerideal apparatus positionsed during each sampling regime.
  6. table 2, total sample total sample reported did not equal the number 5,325. 
  7. Line 180 refered to table 3 however there was no table 3 but table 5 included in the  paper.
  8. Overall, authors may want to revist the  work  presented to improve the structure, grammer and in some instance presnetation quality.

Author Response

Response to Reviewer 2 Comments

Point 1: The abstract need closer attention. In most cases, there are no punctuation marks where they are needed.  In addition it will be good to guide your reader around key results as well as the impleication of the study findings. 

Response 1: Thank you. We added punctuation marks where they are needed and key results as well as the impleication of the study findings:

Of all the tested samples, 1159 (21.8%) were positive, indicating the presence of microorganisms in the tested sample. Species potentially pathogenic for hospital patients were identified, constituting 20.8% of all positive samples (4.6% of all samples). Significantly higher proportion of microorganisms potentially dangerous to patients were isolated from sanitary facilities. Due to the potentially pathogenic microorganisms detected in the tested samples, the authors suggest that in the analyzed hospital the areas requiring a specific level of microbiological purity should be designated and de-scribed, with [specifically] defined cleaning and disinfection protocols.

Point 2: line 40-41 .......opened many "gates" for microorganisms.  The statement seem not to be complete. Are you reffering to their presence in the hosipital premises?

Response 2: Thank you. Yes, that's what we meant. We have changed the statement to the following:

The widespread use of modern treatment techniques (e.g. endoscopy, vascular cannu-lation, assisted respiration, organ transplants, etc.) has resulted in the opening of nu-merous "gateways" for infection by microorganisms among patients treated in hospital settings.

Point 3: Line 42-43  there is a jump.  In my opinion, you needed link their presence with resistence before the justification of their resistence made herein.

Response 3: Thank you. We have changed the statement to the following:

Moreover, the widespread and not always controlled use of antibiotics became one of the reasons for the rise of antimicrobial resistance in hospital environments

Point 4: Check punctuation and the use of capital letters at the start of next sentence (i.e. line 53-54).

Response 4: Thank you. We checked punctuation and the use of capital letters at the start of next sentence in all article, especially line 53-54:

One British study found that a hospital-acquired infection prolongs the stay of a pa-tients in hospital by 250%, nearly tripling the cost of hospital treatment, while after discharge such patients require more frequent services from family doctors and nurses and higher drug spendingg than patients without nosocomial infections.

Point 5: line 109-111.  Considering the longitudinal nature of the study (2012-2018) it will have been much better if the authors detailed  the frequency of the sampling done i.e. once/wk , twice/wk, fortnightly etc  and what elevation were the plates and aerideal apparatus positionsed during each sampling regime.

Response 5: Thank you. We added information about the frequency of the sampling done (line 114). The material was collected once a week. We also added the information about what elevation were the plates and aerideal apparatus positionsed during each sampling regime. It was at the ground level (line 121).

Point 6: table 2, total sample total sample reported did not equal the number 5,325.

Response 6: Thank you. Table 2 shows only the results for samples where bacteria were detected. The samples in which the presence of fungi was detected were not included. We added those explanation in line 151-152.

Point 7: Line 180 refered to table 3 however there was no table 3 but table 5 included in the  paper.

Response 7: Thank you. Table 3 designation was changed to correct. Currently it is table 7.

Point 8: Overall, authors may want to revist the  work  presented to improve the structure, grammer and in some instance presnetation quality.

Response 8: Thank you. We have conducted an intensive rebuild the content of the manuscript.

We have added the results of additional analyzes that present tables 3 to 6, i.e .:

Table 3. Sampling site distribution by the type of the microorganism.

Table 4. Distribution of a sampling location category among areobes, anaeobes and relative anaerobes.

Table 5. Distribution of bacteria families by the sampling area.

Table 6. Distribution microorganism type by the sampling area.

We made also extensive editing of English language and style required I our article.

Reviewer 3 Report

The problem of hospital (nosocomial) infections is very actual even though it is monitored over hundred years. this papers brings credible and reliable results due to the length of research survey. It is worth contributing to this issue worldwide. 

Even though the paper is more microbiology focused than on Nursing, the language is clear, easy to understand.  This topic it sel fis not original, the originality in this paper is the lenght of survay and number od samples and also comparission of pthology and non- pathology microorganism . The results cover one area of the country where the research was done. There for i tis benefitial for the region and i tis inpirational for othe researches - nurses with its methodology - sample acquiring. The main question is what is the incidence of infectious microorganism in hospitalas. This is one very intensive quality indicator all over the world. The paper doesn´t say how many or the percentage of result were possiblly influenced by infected person hospitalizen in the room where theagar plates were placed. The structure of the articel suits the publishers requirements. The number of reference is sufficient, and useful, they underpin the text well.

Please add the citation into the part on lines 89-103 where itś missing.

Author Response

Response to Reviewer 3 Comments

Point 1: The problem of hospital (nosocomial) infections is very actual even though it is monitored over hundred years. this papers brings credible and reliable results due to the length of research survey. It is worth contributing to this issue worldwide.

Even though the paper is more microbiology focused than on Nursing, the language is clear, easy to understand.  This topic it sel fis not original, the originality in this paper is the lenght of survay and number od samples and also comparission of pthology and non- pathology microorganism . The results cover one area of the country where the research was done. There for i tis benefitial for the region and i tis inpirational for othe researches - nurses with its methodology - sample acquiring. The main question is what is the incidence of infectious microorganism in hospitalas. This is one very intensive quality indicator all over the world. The paper doesn´t say how many or the percentage of result were possiblly influenced by infected person hospitalizen in the room where theagar plates were placed. The structure of the articel suits the publishers requirements. The number of reference is sufficient, and useful, they underpin the text well.

Response 1: Thank you. We have added information about incidence of infectious microorganism in hospitals. It is estimated that the incidence of hospital-related infections in Poland is 5.9%. (line 208-209).

We did not conduct research on how many or the percentage of result were possibly influenced by infected person hospitalizen in the room where the agar plates were placed. The aim of the study was to assess the microbiological purity of the hospital environment through the quantitative and qualitative analysis of microbiological contamination of air and surfaces in inpatient treatment using the example of a large clinical hospital in Poland.

Point 2: Please add the citation into the part on lines 89-103 where its missing.

Response 2: Thank you. We added citation on lines 89-103:

Dancer SJ, White LF, Lamb J, Girvan EK, Robertson C. Measuring the effect of enhanced cleaning in a UK hospital: a pro-spective cross-over study. BMC Med 2009;7:28.

Mulvey D, Redding P, Robertson C et al. Finding a benchmark for monitoring hospital cleanliness. J Hosp Infect 2011;77(1):25– 30.

Round 2

Reviewer 1 Report

Line 153 to 156: In the method section, the statements "Statistical analysis of the significance of differences and correlations was carried out 153 in order to investigate the relationship between selected groups of data. Using the Statistica 154 statistical analysis software program, the sampling sites were analyzed in groups accord-155 ing to the cleanliness category and the types of microorganisms present in the examined 156 hospital environment (p<0.05)." stated statistical analysis was done with a software but there is no presentation of the summary of result from it in the "result" section. p<0.05 was put below all the Tables without link inside the Table. Where is the result of significance of differences and correlations was carried out.

The whole conclusion need redrafting for more clarity and better thought flow. The conclusion should be short summary of the observation, contribution to the body of knowledge and recommendation of the study.

Author Response

Point 1: Line 153 to 156: In the method section, the statements "Statistical analysis of the significance of differences and correlations was carried out 153 in order to investigate the relationship between selected groups of data. Using the Statistica 154 statistical analysis software program, the sampling sites were analyzed in groups accord-155 ing to the cleanliness category and the types of microorganisms present in the examined 156 hospital environment (p<0.05)." stated statistical analysis was done with a software but there is no presentation of the summary of result from it in the "result" section. p<0.05 was put below all the Tables without link inside the Table. Where is the result of significance of differences and correlations was carried out.

Response 1: Thank you. We added additional indicators in the text in the results section, such as the chi square of Pearsons and V Kramer (line 168 and 174).

Point 2: The whole conclusion need redrafting for more clarity and better thought flow. The conclusion should be short summary of the observation, contribution to the body of knowledge and recommendation of the study.

Response 2: Thank you. We changed the whole conclusion section to a more clarity and specific. We added a short summary of the observation:

In a humid environment and on damp surfaces (sanitary equipment), gram-negative bacteria of the genera Escherichia, Klebsiella, Pseudomonas and Stenotrophomonas predominate. These can all cause infections in hospitalized patients. The potentially pathogenic species of Staphylococcus aureus is also present.

The presence of potentially pathogenic microorganisms indicate that in the hos-pital we studied, areas which require a specific level of microbiological purity should be designated and described, and specific protocols for cleaning and disinfection should be laid down for those areas.

Reviewer 2 Report

Many thanks for time taken to improve the quality of the paper.

Going through the paper, I find the conclusion especially lines 294-299 a bit confusing.  You mentioned earlier  ..."mainly non-pathogenic microorganisms" were presence and the next sentence stated  that potential pathogenic organism to patient were also detected".  In my opinion there is the need to provide clarity on these statements for better reporting. 

Author Response

Point 1: Going through the paper, I find the conclusion especially lines 294-299 a bit confusing.  You mentioned earlier  ..."mainly non-pathogenic microorganisms" were presence and the next sentence stated  that potential pathogenic organism to patient were also detected".  In my opinion there is the need to provide clarity on these statements for better reporting.

Response 1: Thank you. We changed the whole conclusion section to a more clarity and specific, especially lines 294-299 as following:

In a humid environment and on damp surfaces (sanitary equipment), gram-negative bacteria of the genera Escherichia, Klebsiella, Pseudomonas and Stenotrophomonas predominate. These can all cause infections in hospitalized patients. The potentially pathogenic species of Staphylococcus aureus is also present.

The presence of potentially pathogenic microorganisms indicate that in the hos-pital we studied, areas which require a specific level of microbiological purity should be designated and described, and specific protocols for cleaning and disinfection should be laid down for those areas.